# Analysis of human acetylation stoichiometry defines mechanistic constraints on protein regulation

Bogi Karbech Hansen[1], Rajat Gupta[1], Linda Baldus[2,3], David Lyon[4], Takeo Narita [1], Michael Lammers[2,3], Chunaram Choudhary [1] & Brian T. Weinert[1]

Lysine acetylation is a reversible posttranslational modification that occurs at thousands of sites on human proteins. However, the stoichiometry of acetylation remains poorly characterized, and is important for understanding acetylation-dependent mechanisms of protein regulation. Here we provide accurate, validated measurements of acetylation stoichiometry at 6829 sites on 2535 proteins in human cervical cancer (HeLa) cells. Most acetylation occurs at very low stoichiometry (median 0.02%), whereas high stoichiometry acetylation (>1%) occurs on nuclear proteins involved in gene transcription and on acetyltransferases. Analysis of acetylation copy numbers show that histones harbor the majority of acetylated lysine residues in human cells. Class I deacetylases target a greater proportion of high stoichiometry acetylation compared to SIRT1 and HDAC6. The acetyltransferases CBP and p300 catalyze a majority (65%) of high stoichiometry acetylation. This resource dataset provides valuable information for evaluating the impact of individual acetylation sites on protein function and for building accurate mechanistic models.

[1] Department of Proteomics, The Novo Nordisk Foundation Center for Protein Research, Faculty of Health and Medical Sciences, University of Copenhagen, Blegdamsvej 3B, DK-2200 Copenhagen, Denmark. [2] Institute of Biochemistry, Synthetic and Structural Biochemistry, University of Greifswald, Felix-Hausdorff-Str. 4, Greifswald 17487, Germany. [3] Institute for Genetics and Cologne Excellence Cluster on Cellular Stress Responses in Aging-Associated Diseases, CECAD, University of Cologne, Joseph-Stelzmann-Str. 26, 50931 Cologne, Germany. [4] Disease Systems Biology Program, The Novo Nordisk Foundation Center for Protein Research, Faculty of Health and Medical Sciences, University of Copenhagen, Blegdamsvej 3B, DK-2200 Copenhagen, Denmark. Correspondence and requests for materials should be addressed to C.C. (email: chuna.choudhary@cpr.ku.dk) or to B.T.W. (email: brian.weinert@gmail.com)

Lysine N-ε-acetylation is a reversible protein posttranslational modification (PTM) that was first identified on histones[1]. In the past decade, sensitive mass spectrometry (MS) techniques enabled identification of thousands of acetylation sites on diverse cellular proteins[2–4]. Acetylation can be enzymatically catalyzed by lysine acetyltransferases, however, recent data indicates that acetylation also arises from nonenzymatic reaction with acetyl-CoA[5,6]. Nonenzymatic acetylation potentially targets any solvent accessible lysine residue, suggesting that nonenzymatic acetylation sites are likely to greatly outnumber acetyltransferase-catalyzed sites. As a result, enzyme-catalyzed acetylation is easily overlooked within a vast background of nonenzymatic acetylation, presenting a needle-in-a-haystack problem for identifying these sites. Proteome-wide analyses of lysine acetylation should focus on identifying parameters that will help prioritize the functional relevance of individual sites and provide mechanistic insights. These parameters include regulation by acetyltransferases and deacetylases, dynamic turnover rates, and the stoichiometry of modification. Regardless of the origin of acetylation, enzyme-catalyzed or nonenzymatic, understanding the stoichiometry of modification is important for determining the impact of acetylation on protein function and for building accurate mechanistic models.

We developed a quantitative proteomics method to determine acetylation stoichiometry at thousands of sites by measuring differences in the abundance of native and chemically acetylated peptides[6,7]. We subsequently refined our method by incorporating strict criteria for accurate quantification of acetylated peptides[8]. However, the stoichiometry of acetylation in human cells remains poorly characterized.

Here we determine acetylation stoichiometry at thousands of sites in human cervical cancer (HeLa) cells. We validate our results using known quantities of peptide standards, using recombinant acetylated proteins, and by comparison with acetylated peptide intensity. This high-confidence dataset is used to calculate acetylation copy numbers in cells, to explore the relationship between stoichiometry and regulation by acetyltransferases and deacetylases, and to reveal mechanistic constraints on protein regulation by acetylation.

## Results

**Measuring acetylation stoichiometry.** We measured acetylation stoichiometry in HeLa cells using partial chemical acetylation and serial dilution SILAC (SD-SILAC) to ensure quantification accuracy[8] (Fig. 1a). Two independent biological replicates were performed, each using a different degree of chemical acetylation and inverting the SILAC labeling between experiments. The degree of chemical acetylation was estimated based on the median reduction of unmodified peptides generated by tryptic cleavage at one or two lysine residues (Supplementary Figure 1a). Based on the estimated degree of chemical acetylation, we performed a serial dilution of the chemically acetylated peptides to give median ~1%, ~0.1%, and ~0.01% chemical acetylation. Acetylated peptides were enriched and the differences between native acetylated and chemically acetylated peptides quantified by MS (Supplementary Data 1a). To ensure accurate quantification, we required that the abundance of native acetylated peptides was quantified by comparison with at least two different concentrations of chemically acetylated peptides, and that the measured SILAC ratios agreed with the serial dilution series. SILAC ratios that did not follow the dilution series (allowing up to two-fold variability) were defined as being inaccurately quantified, even though one of the measurements may be correct. Quantification error was reduced when the concentration of chemically acetylated peptides was most similar to native acetylated peptides

(Fig. 1b). However, quantification error was substantially higher than in our previous experiments in bacteria[8], likely due to the greater complexity of the human proteome. The high error rates highlight the need to control for quantification accuracy, and show that comparing native acetylated peptides to just 1% chemically acetylated peptides results in a majority of false quantification (Fig. 1b). The measured stoichiometry of acetylated peptides was significantly and highly correlated between independent experimental replicates (Fig. 1c). The precision of our measurements was also highly reproducible; the median ratio of stoichiometry between replicates was 0.95, and 90% of the measurements varied by less than a factor of two between replicates (Fig. 1d).

For high stoichiometry (>10%) acetylation, the difference between native acetylated and chemically acetylated peptides becomes too small to accurately measure by SILAC quantification, which is typically limited to differences greater than 2-fold in magnitude. At 5% partial chemical acetylation, a peptide with 90% stoichiometry will have a SILAC ratio of 1.006, and a peptide with 50% stoichiometry will have a SILAC ratio of 1.05 (Supplementary Figure 1b). These differences are too small to accurately resolve, and can result in inaccurate stoichiometry measurements that are out of bounds (greater than 100% or negative). Due to the inherent limitations in calculating stoichiometry for these peptides, we set a cutoff of maximum 10% stoichiometry, and classified all sites exceeding this cutoff as having >10% stoichiometry. Only 16 peptides met these criteria, and they harbored previously known high stoichiometry acetylation sites (Supplementary Data 1a), including; histones H3 K23 (2 different peptides), H3 K14, and H4 K16. Other peptides harbored sites on acetyltransferases, such as CBP K1583 (2 different peptides) and N-alpha-acetyltransferase 50 K33, or on proteins that catalyze reactions using acyl-CoAs, such as hydroxymethylglutaryl-CoA lyase K48 and dihydroxyacetone phosphate acyltransferase K643. These data show that, although we are unable to accurately measure high stoichiometry acetylation, we were able to identify these peptides, they represent known or probable high stoichiometry acetylation sites, and they constitute a small portion (0.2%) of the peptides analyzed.

**Validating stoichiometry measurements.** We used known quantities of unmodified and acetylated peptide standards (AQUA peptides) to determine stoichiometry directly at ten sites on three proteins, cortactin (CTTN), nucleolin (NCL), and N-acetyltransferase 10 (NAT10) (Supplementary Data 1b). Stoichiometry determined by partial chemical acetylation (PCA) was significantly correlated ($r = 0.94$) with stoichiometry determined using AQUA peptides (Fig. 1e). Furthermore, stoichiometry measurements differed by a factor of two or less for a majority (7/10) of the analyzed peptides (Fig. 1f). Three sites showed 3.4-, 6-, and 7-fold higher stoichiometry by PCA, indicating overestimation of stoichiometry by PCA or underestimation by AQUA. These differences occurred at the site-level and were therefore not attributable to errors in protein quantification. We think that the agreement between these two methods is notable when considering all possible sources of variability in each measurement.

We further validated our measurements using two recombinant, site-specifically acetylated proteins; malate dehydrogenase (MDH2) K239ac and hydroxymethylglutaryl-CoA lyase (HMGCL) K48ac. Recombinant acetylated proteins (SILAC-light-labeled) were used as a spike-in standard to measure acetylation stoichiometry in SILAC-heavy-labeled HeLa lysate. Stoichiometry measured using two different concentrations of recombinant acetylated protein (spike-in) agreed with our

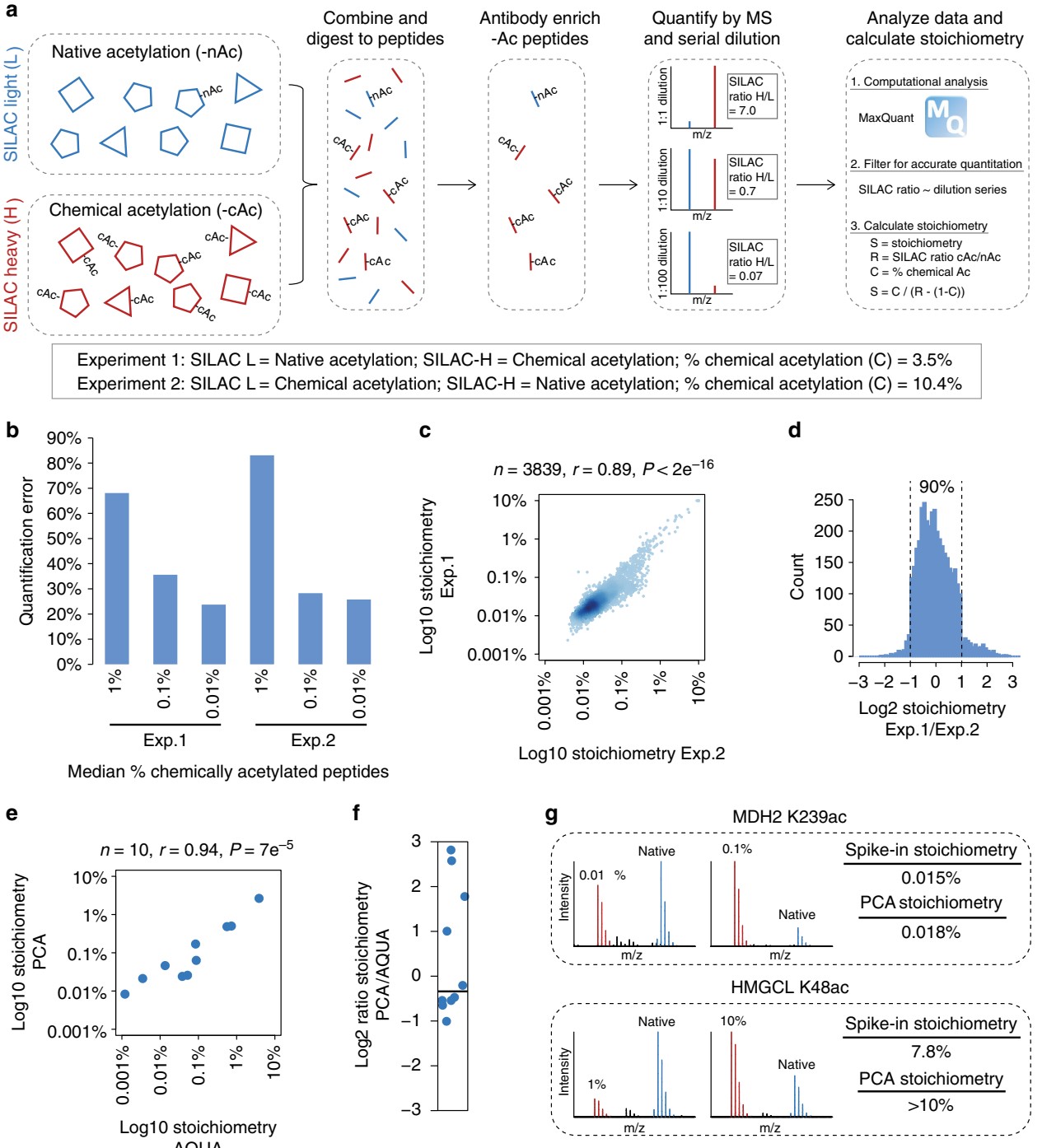

**Fig. 1** Measuring acetylation stoichiometry. **a** Diagram of the method used to measure acetylation (Ac) stoichiometry. **b** The degree of quantification error as determined by the fraction of SILAC ratios at each concentration of chemically acetylated peptides that was not consistent with SILAC ratios measured in at least one different concentration of chemically acetylated peptides. **c** The correlation between stoichiometry measured in independent experimental replicates. The number of peptides ($n$), Pearson's correlation ($r$), and $P$-value ($P$) of correlation are shown. **d** Low absolute variability between experimental replicates. The histogram shows the distribution of Log2 ratios of stoichiometry in Experiment 1/Experiment 2 (Exp.1/Exp.2). **e** The correlation between stoichiometry measured using partial chemical acetylation (PCA) and absolute quantification (AQUA) peptide standards. **f** Low absolute variability between stoichiometry measurements made by PCA and AQUA. **g** Validation of stoichiometry measurements using recombinant acetylated (100%) proteins as a spike-in standard. Stoichiometry was measured at two different concentrations of spike-in protein (SILAC light, red) compared to SILAC heavy-labeled HeLa (blue) for each acetylation site. Source data are provided as a Source Data file

measurements using PCA, further supporting the accuracy of our stoichiometry dataset (Fig. 1g).

Stoichiometry measurements were additionally validated by comparison to acetylated peptide intensity that was corrected for differences in protein abundance. We previously showed that abundance-corrected intensity (ACI) is correlated to acetylation stoichiometry in yeast[6]. ACI was significantly correlated with acetylation stoichiometry in HeLa cells (Supplementary

Figure 1c), however, the predictive power of this correlation was modest ($r = 0.48$–$0.52$). There are several reasons for this modest correlation. Firstly, peptide intensity is inherently variable. Secondly, protein abundance estimates may be inaccurate. We found that outlier data points using iBAQ-based protein abundance were not outliers when using copy-number-based protein abundance (Supplementary Figure 1c), indicating that variability in protein abundance measurements contributes to disagreement between ACI and stoichiometry measurements. Thirdly, antibody-based acetylated peptide enrichment may be peptide-sequence biased, which will introduce further variability. Regardless of these limitations, ACI provides an easy method to estimate the relative stoichiometry of acetylation sites, and the significant correlation with our stoichiometry measurements by PCA provides further support for the accuracy of our measurements.

**Copy number limits the detection of acetylated peptides**. The detection of acetylated peptides is biased to abundant proteins (Fig. 2a). Furthermore, the fraction of lysines that are detected as acetylated on any given protein is significantly correlated with protein abundance (Fig. 2b). This bias is found in every acetylome dataset that we have examined (Supplementary Figure 2a)[9–12], and indicates that acetylation occurs on most lysine residues in cells and that protein abundance is a limiting factor in the detection of acetylated peptides. These data further support the notion that all solvent accessible lysine residues are acetylated to some degree, either enzymatically or nonenzymatically.

Deep proteome measurements detect unmodified peptides from proteins whose abundance spans seven orders of magnitude (Fig. 2a)[13]. This raises the question of why we detect so few acetylated peptides without antibody enrichment (Fig. 2c). The signal intensity of acetylated peptides is comparable to unmodified peptides in the mass spectrometer (Supplementary Figure 2b, c), indicating that we should be able to detect acetylated peptides as readily as unmodified peptides. We compared copy numbers for unmodified peptides to copy numbers for acetylated peptides

as determined from our stoichiometry measurements. The distribution of acetylated peptide copy number shows that, in the absence of acetylated peptide enrichment, most acetylated peptides are at or below the detection limit of the mass spectrometer, even in deep proteome measurements (Fig. 2d). In contrast, acetylated peptides that were detected without antibody enrichment occurred at copy numbers that were within the detectable range of unmodified peptides (Fig. 2e). Thus, our stoichiometry measurements are consistent with the inability to detect acetylated peptides without enrichment. Strikingly, our data indicate that some acetylation events are so rare that they occur at a copy number that is less than one per cell (Fig. 2d).

**Properties of high stoichiometry acetylation**. We measured the stoichiometry of acetylated peptides; however, individual acetylation sites may occur on multiple different peptides due to incomplete tryptic digestion, protein N-terminal acetylation, or oxidized methionine residues. To examine acetylation stoichiometry at the site-level, we calculated the summed stoichiometry of peptides containing the same acetylation site (Supplementary Data 1c). This resulted in stoichiometry measurements for 6829 sites, with a median stoichiometry of just 0.02% (1/4000 molecules) (Fig. 3a). This represents very low levels of acetylation for most sites, only 1% (66 sites) displayed stoichiometry >1%, and ~15% (1014 sites) displayed stoichiometry >0.1%.

We performed UniProt keyword enrichment analysis to examine the functional categories of proteins that are associated with higher stoichiometry (>0.23% or >1%) acetylation (Fig. 3b). Higher stoichiometry acetylation was overrepresented on nuclear proteins involved in chromatin regulation and transcription. This observation is consistent with the known nuclear functions of acetyltransferases, deacetylases, and acetylated lysine-binding bromodomain proteins. In fact, the keywords Bromodomain and Acetyltransferase were significantly enriched in the group of proteins with high stoichiometry acetylation. We were unable to calculate stoichiometry for doubly acetylated peptides because of the low frequency of chemical acetylation at both positions.

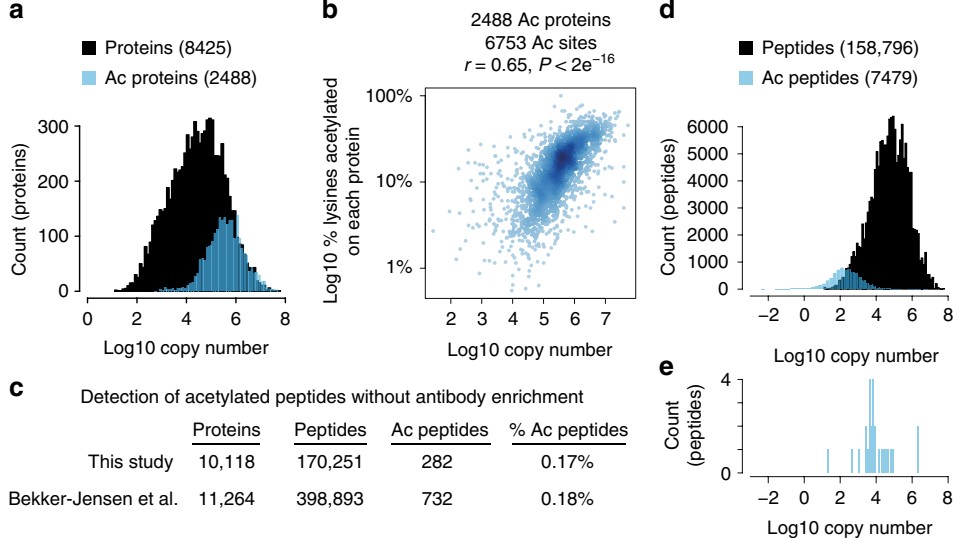

**Fig. 2** Stoichiometry limits the detection of acetylated peptides. **a** Acetylation is biased to detection on abundant proteins. Protein copy number estimates are from[42]. **b** The fraction of acetylated lysines detected on any given protein is correlated with protein abundance. The scatterplot shows the % lysines acetylated and copy numbers of 2488 acetylated proteins containing 6753 acetylation sites. The Pearson's correlation (r), and P-value (P) of correlation are shown. **c** The number of peptides and acetylated peptides (Ac peptides) detected in deep proteome measurements from this study and[13]. **d** The distribution of peptide copy numbers from a deep proteome measurement and acetylated peptide copy numbers calculated from the peptide stoichiometry and protein copy number. **e** The distribution of acetylated peptide copy numbers for acetylated peptides that were detected without prior antibody enrichment. Source data are provided as a Source Data file

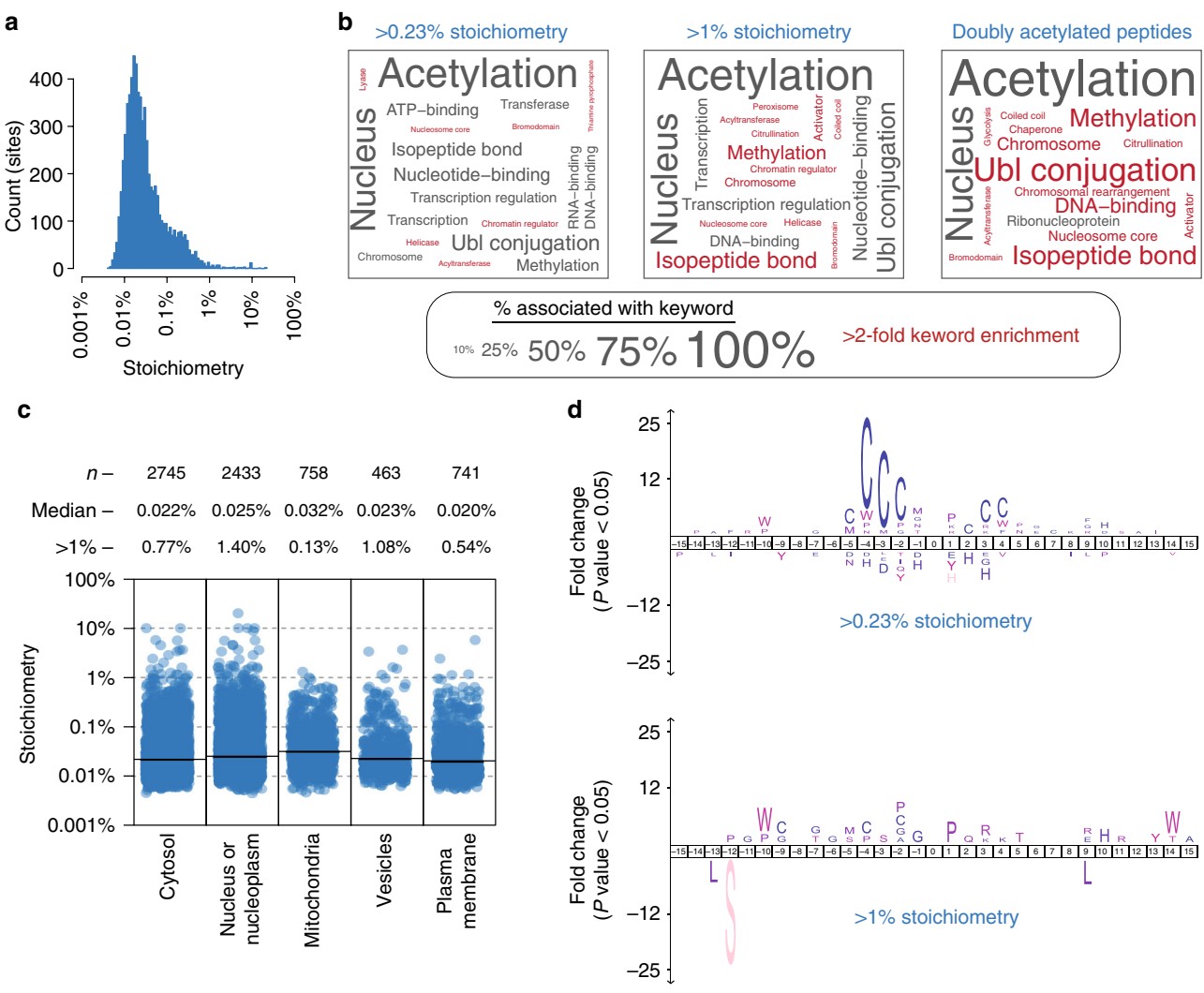

**Fig. 3** Properties of high stoichiometry acetylation. **a** The distribution of acetylation site stoichiometry for the 6829 sites measured in this study. **b** UniProt keyword enrichment for the indicated classes of high stoichiometry acetylation sites (>0.23% and >0.1%) and for doubly acetylated peptides. The size of the text is related to the fraction of sites associated with the keyword, and keywords that were more than two-fold enriched are colored red. **c** Subcellular compartment analysis based on the Human Protein Atlas[14]. Category scatterplots show the distributions of acetylation site stoichiometry in each subcellular compartment. The number (n) of sites analyzed, median stoichiometry (median), and percentage of sites with >1% stoichiometry (>1%) is shown. **d** Amino acid sequence logos of the indicated classes of acetylation sites using IceLogo[15]. Source data are provided as a Source Data file

However, given the low stoichiometry of acetylation, doubly acetylated peptides are unlikely to occur by random chance and most likely reflect the activity of acetyltransferases. Consistent with this idea, doubly acetylated peptides occurred on proteins that were overrepresented for the same UniProt keywords that were associated with high stoichiometry acetylation (Fig. 3b). Thus, sites occurring on doubly acetylated peptides may occur at high stoichiometry and are likely to be enzyme-catalyzed.

To investigate the relationship between stoichiometry and subcellular compartmentalization we used immunofluorescence-based protein localization as determined by the Human Protein Atlas[14]. Acetylation stoichiometry was broadly distributed and mostly similar in every subcellular compartment analyzed (Fig. 3c). Mitochondrial acetylation occurred at a slightly, yet significantly ($P < 5e^{-5}$, Wilcoxon test), higher median stoichiometry. However, mitochondria contained the smallest fraction of high (>1%) stoichiometry acetylation sites. In contrast, the nucleus contained the greatest fraction of high stoichiometry sites, which was approximately an order of magnitude greater than in mitochondria (Fig. 3c).

We used IceLogo[15] to determine whether high stoichiometry acetylation was associated with neighboring amino acids. Cysteine residues were notably overrepresented for sites with >0.23% stoichiometry (10-fold higher than median stoichiometry), particularly in the −4, −3, and −2 positions (Fig. 3d). However, this bias was absent when examining sites with >1% stoichiometry, indicating that this overrepresentation was associated with sites with moderately elevated stoichiometry. Remarkably, sites with cysteine residues in the −4, −3, or −2 position constituted 35% (159/460) of the sites with >0.23% stoichiometry. UniProt keyword enrichment analysis of the proteins harboring these sites found a variety of enriched keywords (Supplementary Data 1d). However, unlike high stoichiometry acetylation in general (Fig. 3b), keywords describing processes associated with nuclear acetyltransferases, such as Nucleus, Transcription, and Chromosome were notably absent. These data suggest that cysteine residues may promote nonenzymatic acetylation of downstream lysine residues, and these sites constitute a substantial portion (35%) of sites with an elevated (>0.23%) stoichiometry of acetylation. This conclusion is

supported by a recent study that uncovered a similar bias for higher stoichiometry acetylation at sites with proximal cysteine residues[16].

**Histone acetylation**. We measured stoichiometry at 57 histone sites and found that high stoichiometry acetylation was mostly restricted to sites on the N-terminal tails of core histones H2B, H3, and H4 (sites on H2A were not measured) (Fig. 4a). The stoichiometry of histone H3 and H4 acetylation sites has been extensively studied[17–20], and our measurements are comparable to these previous measurements (Fig. 4b). However, our method does not measure the stoichiometry of doubly acetylated peptides that can arise from the lysine-rich N-terminal tails of the core histones H2A, H2B, H3, and H4. Feller et al.[18] found that the stoichiometry of mono-acetylated histones H3 and H4 were more abundant than di-acetylated H3 and H4, with the exception of H4 K5 + K12 and H3 K18 + 23 (which is more abundant than K18

alone, but less abundant than K23 alone). We detected doubly acetylated peptides containing the following sites on H2A (K5 + K9, K9 + K11, K11 + K13), H2B (K5 + K11, K11 + K12, K15 + K16, K16 + K20, K20 + K23, K34 + K43, and K116 + K120), H3 (K9 + K14, K18 + K23, and K27 + K36), and H4 (K8 + K12 and K12 + K16). The stoichiometry of di-acetylated peptides remains unexplored for H2B, and these peptides may be more abundant than their mono-acetylated counterparts used to calculate stoichiometry in this study. In addition, some acetylated peptides from histone tails may not be detected because of their small size. Thus, our estimates of histone acetylation stoichiometry may underestimate the actual native stoichiometry at these positions because these sites also occur on di-acetylated peptides or peptides that we are unable to detect with our methodology. Regardless, our data suggest that the stoichiometry of H2B acetylation ranges from 0.5% to 5.6%. N-terminal H2B sites are primarily acetylated by the CBP/p300 acetyltransferases[12], which also target H3K27 and H3K36. N-terminal H2B sites (K5, K11,

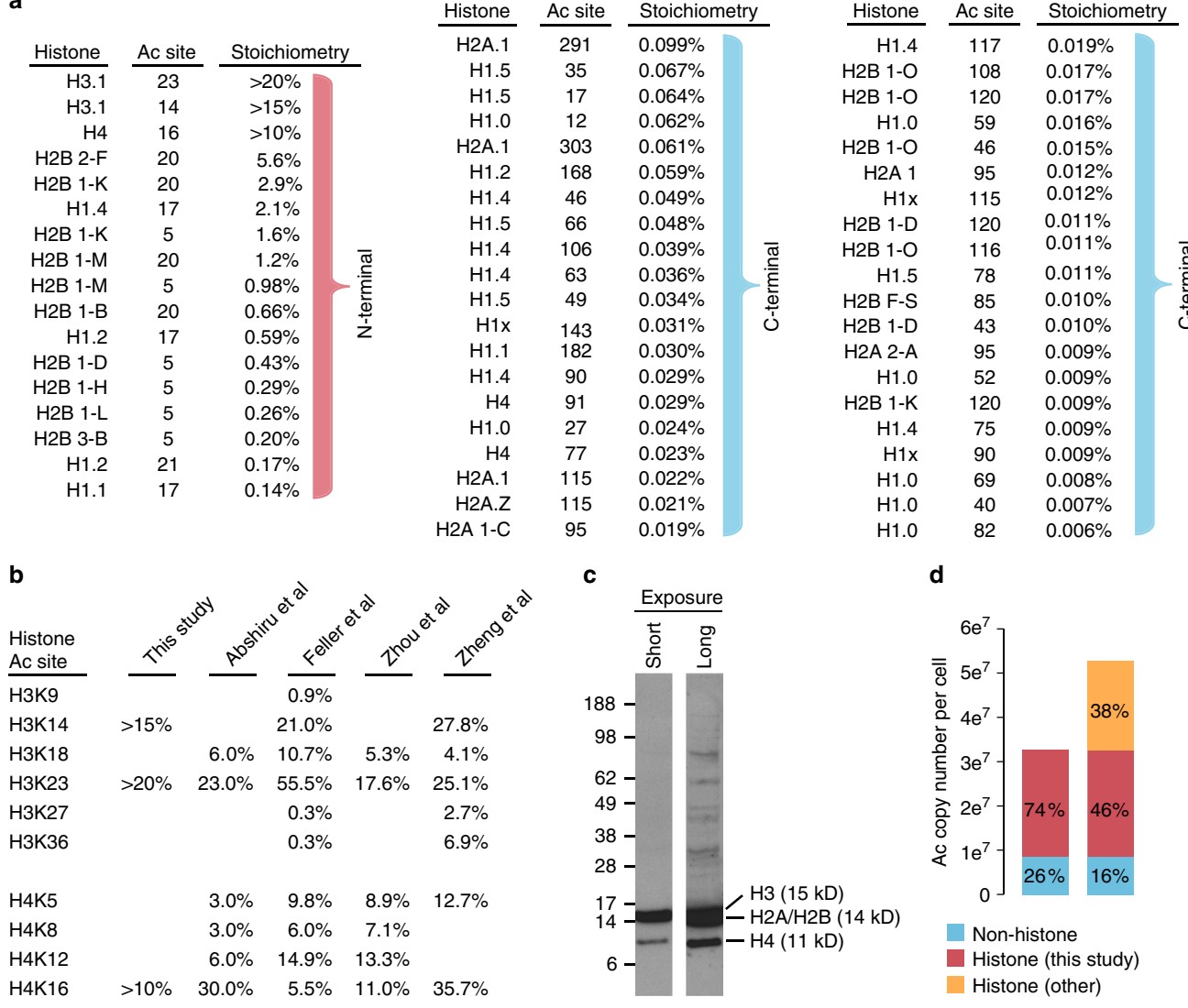

**Fig. 4** Stoichiometry of histone acetylation. **a** The diagram shows all histone acetylation sites whose stoichiometry was determined in this study. The sites are ordered by descending stoichiometry. Note that high stoichiometry sites occur on the N-termini of core histones. **b** The stoichiometry of histone acetylation sites as determined in four independent studies[17–20]. **c** An anti-acetylated lysine immunoblot of HeLa whole cell lysate. Cells were boiled in 2% LDS to ensure histone extraction. Histones are annotated based on their expected molecular weight. **d** Histone acetylation sites constitute a majority of acetylated lysine residues in cells. Stoichiometry and protein copy numbers were used to calculate the number of acetylated lysine residues for the indicated classes of proteins. Source data are provided as a Source Data file

K12, K15, and K16) show faster deacetylation kinetics compared to K20[12]. Interestingly, the stoichiometry of K20 is greater than K5, indicating that the lower stoichiometry at K5 is possibly due to its more rapid turnover.

Histones are some of the most abundant and most highly acetylated proteins in cells. Anti-acetylated lysine antibodies prominently detect histone acetylation in western blots of whole cell lysates, suggesting that histones harbor most of the acetylated lysine residues in cells (Fig. 4c). Using our acetylated lysine copy number estimates we found that histone acetylation accounted for 74% of the acetylated lysine residues in cells (Fig. 4d). If we include histone sites whose stoichiometry was measured by independent studies[17–20], the fraction of acetylated lysines occurring on histones increases to 84% (Fig. 4d). These estimates do not account for several sites on H2B (K11, K12, K15, K16, and K23), as well as sites on H2A (K4, K5, K7, K9, K11, and K13). Thus, histone acetylation likely accounts for an even greater proportion of the acetylated lysine residues found in cells.

**Regulation by deacetylases.** We analyzed the stoichiometry of lysine deacetylase (KDAC)-regulated acetylation sites by comparing the data obtained in this study with a previous analysis of deacetylase inhibitors in HeLa cells[21]. Class I KDACs (HDAC 1, 2, 3, and 8) are specifically targeted by the class I inhibitors apicidin, MS-275, valproic acid, and sodium butyrate; the class IIb KDAC HDAC6 is specifically targeted by tubacin; and nicotinamide inhibits the activity of $NAD^+$-dependent Sirtuin deacetylases, but mostly affected SIRT1-regulated sites in mammalian cells[21]. To analyze class I KDAC regulated sites, we used the median increased acetylation caused by apicidin, MS-275, valproic acid, and sodium butyrate.

KDAC inhibitors regulated sites with a broad range of stoichiometry (Fig. 5a). Class I KDAC inhibitors regulated substantially greater proportions of moderately increased (>0.23%) and high stoichiometry (>1%) acetylation compared to tubacin or nicotinamide. The stoichiometry of class I regulated sites was significantly higher than not-regulated (NR) sites, while

the distributions of tubacin and nicotinamide regulated sites was not significantly different than NR sites. Furthermore, sites that were most sensitive to KDAC inhibitors (>4× increased acetylation) showed an increasing proportion of higher stoichiometry sites for the class I inhibitors, but stayed the same or decreased for tubacin and nicotinamide (Fig. 5a). Thus, while tubacin and nicotinamide regulate a greater portion of the acetylation sites quantified in this study (8.6% and 8.8%, respectively) than class I inhibitors (2.6%), the class I inhibitors regulate a greater proportion of higher stoichiometry acetylation.

**Regulation by the CBP and p300 acetyltransferases.** The homologous Creb-binding protein (CBP)/E1A-binding protein p300 (p300) acetyltransferases are important regulators of cell-type-specific and signaling-regulated gene expression[22]. CBP/p300 acetyltransferase activity is essential for promoting gene transcription and CBP/p300 targets a large proportion of the acetylome[12]. CBP/p300-regulated sites constituted 12.7% of the sites analyzed in this study, indicating that CBP/p300 targets more than one out of every ten acetylation sites. CBP/p300 targeted a similar proportion (11.5%) of low stoichiometry ( < 0.2%) sites, and an increasing proportion of higher stoichiometry sites, up to 65% of the sites with stoichiometry exceeding 1% (Fig. 5b). Thus, CBP/p300 acetylates a majority of high (>1%) stoichiometry acetylation sites. The stoichiometry of CBP/p300-regulated sites also increased with the degree of downregulated acetylation in the absence of CBP/p300 catalytic activity (Fig. 5c), indicating that the sites most affected by loss of CBP/p300 tend to be more highly acetylated.

**Stoichiometry of functionally characterized sites.** The activity and subcellular localization of eukaryotic translation initiation factor 5 A (eIF5A) is regulated by PCAF-catalyzed acetylation at K47[23]. We found that eIF5A was more than 10% acetylated at K47, consistent with a regulatory role for acetylation at this position. DNA methyltransferase 1 (DNMT1) is acetylated by the TIP60 acetyltransferase, and acetylation promotes ubiquitin-

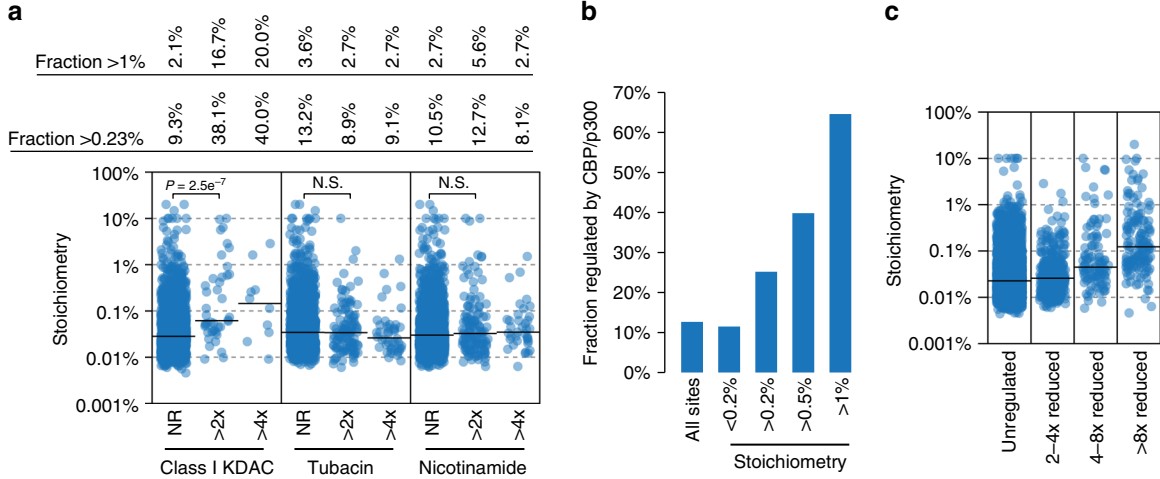

**Fig. 5** Stoichiometry of deacetylase- and CBP/p300-regulated acetylation sites. **a** The category scatterplot shows the distributions of acetylation sites that are not regulated (NR), more than two-fold (>2×) upregulated, or more than four-fold (>4×) upregulated, by the indicated deacetylase inhibitors as determined by[21]. Class I KDAC inhibitors primarily target HDACs 1, 2, 3, and 8, and were determined by the median SILAC ratio of apicidin, MS-275, valproic acid, and sodium butyrate-treated HeLa cells. Tubacin is an HDAC6 inhibitor and nicotinamide inhibits Sirtuin deacetylases, but the regulated sites are mostly attributed to SIRT1[21]. **b** CBP/p300 regulates an increasing fraction of high stoichiometry acetylation sites. CBP/p300-regulated sites were determined by[12]. **c** Acetylation sites that are most affected (>8× reduced) by loss of CBP/p300 activity have higher median stoichiometry than sites that are only modestly affected (2–4× reduced). Source data are provided as a Source Data file

dependent protein turnover[24]. We found that DNMT1 harbored high stoichiometry acetylation at K335 (0.57%) and K675 (0.1%). Pyruvate dehydrogenase E1 alpha 1 subunit (PDHA1) is acetylated at K321 by acetyl-CoA acetyltransferase 1 (ACAT1), which recruits PDH kinase (PDK) to inhibit pyruvate dehydrogenase activity[25]. We found that PDHA K321 was 0.6% acetylated, supporting high stoichiometry acetylation at this position.

Glucose-6-phosphate dehydrogenase (G6PD) is reported to be negatively regulated by KAT9-dependent acetylation of K403, and activated by SIRT1-dependent deacetylation[26]. We found that G6PD is just 0.02% acetylated at K403. Acetylation of phosphoglycerate kinase 1 (PGK1) at K220 disrupts its activity by inhibiting binding to ADP[27]. We found that PGK1 is 0.03% acetylated at K220 in HeLa cells. PCAF is reported to regulate cyclin dependent kinase 2 (CDK2) activity by acetylating CDK2 at K33 in the ATP-binding active site[28]. We found that CDK2 K33 is just 0.05% acetylated, however, CDK1 K33 was 4.5% acetylated, suggesting that acetylation may play a greater role in regulating CDK1 activity[3]. In each of the above examples, acetylation is reported to reduce enzymatic activity, however, the stoichiometry of acetylation suggests that acetylation would need to be dramatically increased (100-fold or more) in order to have a substantial impact on protein function.

Low stoichiometry acetylation does not necessarily indicate a lack of function. Rather, the mechanism-of-action determines whether low stoichiometry acetylation is sufficient to regulate protein function. Acetylation that imparts a gain-of-function or regulates protein activity at a specific time and/or place, could be regulated by low stoichiometry acetylation. For example, acetylation at K220 regulates the activity of microtubule associated protein RP/EB family member 1 (EB1), specifically during mitosis and at spindle microtubule plus ends[29]. Although we found that EB1 K220 was only 0.02% acetylated, the mechanism-of-action is consistent with the low observed stoichiometry of modification.

## Discussion

Here we provide a validated resource of acetylation stoichiometry at thousands of sites in the most widely studied human cell line (HeLa). Our data shows that the vast majority of acetylation occurs at very low stoichiometry. Thus, as a general rule, the mechanisms by which acetylation regulates protein function should agree with a low stoichiometry of modification. There are a large number of metabolic enzymes whose catalytic activity is reduced by site-specific acetylation[30]. Many studies relied on acetylation mimicking glutamine-substitution mutations to assay the impact of acetylation on these proteins, a method that results in 100% stoichiometry of modification. However, our measurements indicate that the vast majority of acetylation occurs at a stoichiometry that is much less than 1%, and is therefore not likely to impact protein activity through a loss-of-function mechanism at a single acetylation site. Low stoichiometry acetylation is compatible with gain-of-function mechanisms, or in processes that are spatially or temporally restricted. However, even for gain-of-function mechanisms such as increased catalytic activity, a high degree of acetylation may be required to have a measurable impact on overall activity. Thus, understanding the stoichiometry of acetylation is important for formulating accurate mechanistic models when evaluating the impact of acetylation on protein function. High stoichiometry sites may be particularly interesting because they are more likely to be enzyme-catalyzed. However, more studies are required to determine if high stoichiometry is a good indicator of functional importance. Enzymatic acetylation does not necessarily indicate a regulatory function, and high stoichiometry acetylation may also occur by nonenzymatic mechanisms.

We previously showed that the Sirtuin deacetylases SIRT3 and CobB suppress acetylation at hundreds of sites to levels that are equal to or less than the median stoichiometry of modification[7,8]. These data support the idea that Sirtuin deacetylases may have a general role suppressing nonenzymatic acetylation to preserve protein function[31]. Here we find that SIRT1 and HDAC6 also suppress acetylation at regulated sites to levels that are comparable to the median stoichiometry of acetylation. SIRT1 likely suppresses the activity of nuclear acetyltransferases, ensuring tight control over the sites targeted by these enzymes. HDAC6 deacetylates a large number of cytoplasmic acetylation sites, most of which are acetylated by unidentified acetyltransferases, or nonenzymatically. Interestingly, the HDAC6 inhibitor Bufexamac sensitizes cells to nonenzymatic acetylation caused by aspirin[32], suggesting that HDAC6 may have a role in protecting cells from nonenzymatic acetylation stress. It is important to note that the activity of these deacetylases as general suppressors of acetylation at hundreds of sites, or as dynamic regulators of individual acetylation sites, is not mutually exclusive. Deacetylases may also target hundreds of acetylation sites for no reason whatsoever; if such activity is not evolutionarily disadvantageous, it will not be selected against.

Our data highlights the outsized impact of CBP and p300 on the acetylome. CBP/p300 targets up to 20% of all acetylation sites in cells, and acetylates proximal proteins in a sequence-independent manner[12]. Here we find that CBP/p300 targets a majority (65%) of high (>1%) stoichiometry acetylation. However, CBP/p300 also acetylated many sites to a low stoichiometry of modification. These low stoichiometry sites likely include both functionally important regulatory acetylation and non-functional, off-target acetylation. This presents a challenge to prioritizing sites for mechanistic analyses and suggests that additional parameters, such as dynamic turnover rates[12] and conditional regulation, are needed to identify potentially interesting sites.

Several other groups have also used chemical acetylation to measure acetylation stoichiometry, often reporting substantially higher stoichiometry than we observed in our experiments[19,33,34]. One crucial difference between these studies and our own is that we use a low degree of chemical acetylation (~5–10%) and dilute the fraction of chemically acetylated peptides to allow for accurate quantification[8]. Other studies used stable-isotope-labeled acetylating agents, such acetic anhydride and NHS-acetate, to completely acetylate all free lysine residues and isotopically label the chemically acetylated fraction at the same time[19,33,34]. While this is an elegant approach, incomplete isotopic labeling of the acetylating agents limits the resolution to stoichiometry greater than 1–2%, and it is not possible to independently dilute the chemically acetylated peptides to ensure accurate quantification. Most of these studies[19,33–35] did not validate their results using orthogonal methods, and the accuracy of their measurements is likely impacted by the limited dynamic range of accurate quantification by mass spectrometry[36,37]. One study found that ~75% of their measurements were impacted by false quantification[19], and we found that ~90% of our measurements were inaccurate when quantifying the differences between completely (100%) acetylated peptides and native acetylated peptides[8]. The differences between 100% acetylated and native acetylated peptides are likely too large to be accurately quantified by MS. Given the differences in acetylation stoichiometry reported by independent studies, care should be taken to carefully validate these measurements.

The paucity of high stoichiometry acetylation detected in our study is somewhat disappointing. However, as we show, it is consistent with the inability to detect acetylated peptides in deep proteome measurements. One possibility is that proteins are deacetylated during protein extraction, resulting in uniformly

decreased acetylation. We consider this unlikely for several reasons. We and others can quantify acetylation changes that occur in vivo, indicating that these differences are preserved during protein extraction. We furthermore compared acetylation after lysing cells by three different methods and found that acetylation levels were unaffected by the method of protein extraction (Supplementary Figure 3). Thus, we believe that our measurements indicate the stoichiometry of acetylation that occurs on proteins inside cells.

## Methods

**Cell culture.** HeLa (ATCC: CCL-2) cells were tested for mycoplasma contamination and grown in DMEM supplemented with 10% FBS, 2 mM L-glutamine, and 1% penicillin/streptomycin. SILAC media was supplemented with arginine and lysine (SILAC Light) or with heavy isotope-labeled arginine ($^{13}C_6$,$^{15}N_4$-arginine, Sigma) and lysine ($^{13}C_6$,$^{15}N_2$-lysine, Cambridge Isotope Laboratories) (SILAC heavy) in media containing dialyzed serum (Sigma). Cells were cultured at 37 °C in a humidified incubator at 5% $CO_2$. At a confluency of ~90%, cells were washed twice with PBS and lysed in ice-cold modified RIPA buffer (50 mM Tris, pH 7.5, 150 mM NaCl, 1 mM EDTA, 1x mini complete protease inhibitor cocktail (Roche), 10 mM nicotinamide, and 5 μM trichostatin A). Lysates were mixed with 1/10 volume of 5 M NaCl to release chromatin-bound proteins and incubated for 15 min on ice. Subsequently, lysates were homogenized by sonication (6 × 10 sec, 15 W), cleared by centrifugation (20,000 × $g$, 15 min, 4 °C), and the supernatant precipitated by addition of four volumes of −20 °C acetone. Precipitates were re-dissolved in 8 M guanidine HCl, 50 mM Hepes pH8.5 and protein concentration was determined by Quick-start Bradford assay (Bio-Rad).

**Chemical acetylation.** Protein lysates in 8 M guanidine HCl were mixed with 1/10 volume 1 M acetyl-phosphate (Sigma) prepared in $H_2O$, and the acetylation reaction was allowed to proceed for 2 h at 37 °C. Control reactions were prepared by mixing with 1/10 volume $H_2O$. The acetyl-phosphate was quenched by diluting the reaction five-fold in 8 M Guanidine HCl, 100 mM Tris pH8. Control and chemically acetylated SILAC-labeled protein lysates were mixed in equal portions according to the protein concentration as determined by Bradford assay (above). The samples were digested (described below) and analyzed by mass spectrometry to determine the degree of partial chemical acetylation (PCA) (Supplementary Figure 1a). The measured SILAC ratios were additionally used to adjust the mixing of SILAC lysates in subsequent experiments.

**Protein digestion.** Protein lysates were reduced and alkalated with 5 mM TCEP and 5 mM chloracetamide for 45 min at room temperature. Approximately 10 mg protein per condition was diluted to 2 M guanidine HCl with 50 mM Hepes, pH8 and digested by endoproteinase Lys C (1:200 w/w; Wako) for 2 h at room temperature. The lysates were further diluted to 1 M guanidine HCl and digested with trypsin protease (1:200 w/w; Sigma-Aldrich) for 16 h at 37 °C. Digestion was stopped by the addition of trifluoroacetic acid (TFA) to a final concentration of 1%. Digests were cleared by centrifugation (2500 × $g$, 5 min) and loaded onto reversed-phase C18 Sep-Pak columns (Waters), pre-equilibrated with 5 ml acetonitrile and 2 × 5 ml 0.1% TFA. Peptides were washed with 0.1% TFA and $H_2O$, and eluted with 50% acetonitrile.

**Acetylated peptide enrichment.** Peptide concentration was determined by UV spectrometry. The first experiment was performed in technical replicates using two different peptide fractionation strategies. In the first strategy, peptides were pre-fractionated by high pH reversed-phase chromatography[11] to 6 fractions followed by acetylated peptide enrichment and micro-scale (in stage-tip) scale strong cation exchange (SCX) chromatography to an additional 3 fractions (pH4.5, 5.5, and 9.0), for a total of 18 fractions. The second strategy was to perform acetylated peptide enrichment on the entire peptide digest, followed by micro-scale SCX into 5 fractions (pH3.5, 4.0, 4.5, 5.5, and 9.0). The second experiment was performed using the second fractionation strategy only. For acetylated peptide enrichment, the peptides were mixed with 100 μl 10x IP buffer (500 mM MOPS; pH 7.2, 100 mM Na-phosphate, 500 mM NaCl, 5% NP-40) per 5 mg peptides. The acetonitrile was removed and the volume reduced to ~1 ml, by vacuum centrifugation. The final volume was adjusted with $H_2O$ to a concentration of 5 mg/ml. 40 μl of anti-acetylated lysine antibody (PTMScan Acetyl-Lysine Motif [Ac-K] Kit, Cell Signaling Technology) was washed 3× in 1 mL IP buffer, the peptides were clarified by centrifugation at 20,000 × $g$ for 5 min, and the peptide supernatant was mixed with the anti-acetylated lysine antibody. Peptides were enriched overnight at 4 °C, washed 3 × in 1 ml cold (4 °C) IP buffer, 4× in 1 ml cold IP buffer without NP-40, and 1× in 1 ml $H_2O$. All wash buffer was removed using a 26 gauge needle on an aspirator. Acetylated peptides were eluted with 100 μl 0.15% TFA, repeated for a total of three times. Peptides were loaded directly onto a micro-SCX column, fractionated as described above, and de-salted on C18 stage-tips[38].

**Mass spectrometry.** Peptides were analyzed by nanoflow liquid chromatography-coupled tandem mass spectrometry (nLC-MS/MS) using a Proxeon easy nLC 1200 connected to a Q-Exactive HF mass spectrometer (Thermo Scientific). The Q-Exactive was operated in profile mode using positive polarity and a Top10 data dependent acquisition (DDA) method with the following settings: Spray voltage = 2 kV, S-lens RF level = 50, heated capillary = 275 °C. Full scan (MS1) was performed with an m/z range of 300–1750 at 60,000 resolution with a target value of $3 × 10^6$ ions and a maximum fill time of 20 milliseconds (ms). Fragment (MS2) scans were performed at a resolution of 15,000 with a target value of $5 × 10^4$ ions, a maximum injection time of 110 ms, an isolation width of 1.3 m/z, a normalized collision energy (NCE) of 28, and a fixed first mass of 100 m/z. Peptide were separated by nanoflow chromatography using an EASY-nLC 1000 system (Thermo Scientific) connected to a 15 cm capillary column packed with 1.9 μm Reprosil-Pur C18 beads (Dr. Maisch). Column temperature was maintained at 40 °C using an integrated column oven (PRSO-V1, Sonation GmbH) Peptides were eluted by a gradient of acetonitrile (ACN) in 0.1% formic acid. A typical run utilized a 120 min gradient followed by 15 min wash and equilibration. A linear gradient at 250 nl/ min of 8-24% ACN (90 min) and 24–40% ACN (30 min) was followed by a wash at 40–80% ACN (5 min) and 80%–8% ACN (5 min), and equilibration at 8% ACN (5 min).

Raw MS data were analyzed using MaxQuant (developer version 1.5.5.4) with the integrated Andromeda search engine[39] to search the UniProt human FASTA (downloaded 6 July 2015). The following Andromeda settings were used; initial search mass tolerance of 20 ppm, main search mass tolerance of 6 ppm for parent ions and 20 ppm for HCD fragment ions, trypsin specificity with a maximum of two missed cleavages, cysteine carbamidomethylation as a fixed modification, and N-acetylation of proteins, oxidized methionine, and acetylated lysine as variable modifications. Acetylated peptides were filtered for a minimum Andromeda score of 40, as per the default settings for modified peptides. Known contaminants were removed based on classification by MaxQuant. The false discovery rate (FDR) was estimated for peptides and proteins individually using a target-decoy approach allowing a maximum of 1% false identifications from a reversed sequence database.

**Calculation of acetylation stoichiometry.** We used MaxQuant to analyze the SILAC ratios of native and chemically acetylated peptides. In order to accurately calculate stoichiometry it is important to compare the SILAC ratios of individual, singly acetylated peptides from the "evidence.txt" table. We do not use the "Acetyl (K)Sites.txt" table since the SILAC ratios of individual sites are sometimes derived from multiple peptides. Likewise, entries in the "modificationSpecificPeptides.txt" table can include different positions of acetylation within the same peptide sequence. Starting with the "evidence.txt" table the following actions were performed;

1. Remove all entries where Reverse = +, Potential contaminant = +, Acetyl (K) = 0, and Acetyl (K) = 2. This removes reverse and contaminant entries and results in only entries containing singly acetylated peptides.
2. The Modified Sequence was used as a unique identifier to calculate the median SILAC ratio and summed peptide intensity for multiple instances of any given acetylated peptide in each experiment.
3. SILAC ratios were tested for agreement with the dilution series, allowing for up to two-fold variability[8]. Stoichiometry was calculated using only peptides that met these strict criteria.
4. Stoichiometry was calculated as follows; Stoichiometry (S), degree partial chemical acetylation (C), dilution factor for acetylated peptides (D), and SILAC ratio partial chemical acetylation/native acetylation (R). $S = (C)/((R*D)-(1-C))$. The dilution factors (D) were as follows: Experiment 1, (~1% D = 4.23), (~0.1% D = 42.3), (~0.01% D = 423), Experiment 2, (~1% D = 6.37), (~0.1% D = 63.7), (~0.01% D = 637). The median degree of partial chemical acetylation (C) was 3.53% and 10.38%, for experiments 1 and 2, respectively, (Supplementary Figure 1a).

The above mentioned .txt files can be found via the PRIDE[40] partner repository with the dataset identifier PXD009994. Relative acetylation stoichiometry was estimated using acetylation site intensity corrected for differences in protein abundance, otherwise referred to as abundance-corrected intensity (ACI). ACI was calculated by dividing acetylation site intensity by iBAQ protein abundance.

**Validation by AQUA and spike-in.** Two different unmodified AQUA peptides (Thermo Scientific, AQUA quant pro) per protein were added to SILAC heavy-labeled HeLa peptides to determine the concentration of each protein (CTTN, NCL, and NAT10). Acetylated AQUA peptides were then added to final stoichiometry of 1%, 0.1%, or 0.01%, in accordance with each individual protein's concentration. Acetylated peptides were enriched as described above and analyzed by mass spectrometry. For validation by recombinant acetylated protein spike-in, recombinant acetylated protein was added to HeLa lysate and digested to determine the concentration of protein present in the lysate. Based on the initial analysis, recombinant HMGCL K48ac was added at a 1:1 (100%), 1:10 (10%), and 1:100 (1%) stoichiometry and MDH2 K239ac was added at a 1:1 (100%), 1:1,000 (0.1%), and 1:10,000 (0.01%) stoichiometry. Acetylated peptides were enriched from the 1:10 and 1:100 dilutions (HMGCL) and the 1:1,000 and 1:10,000 dilutions (MDH2) before analysis by mass spectrometry. The 1:1 dilutions were analyzed to

determine the protein mixing ratio and to correct the stoichiometry measurements accordingly. The mixing ratio of spike-in/HeLa was 1.48 for HMGCL and 0.70 for MDH2.

**Expression and purification of recombinant acetylated proteins**. MDH2 (K239AcK) and HMGCL (K48AcK) were expressed from a pRSF-Duet-vector containing the HMGCL coding region with an amber-stop codon at the position encoding for acetyl-L-lysine incorporation, and additionally the coding regions for a synthetically evolved pair of *Methanosarcina barkeri* MS tRNA$_{CUA}$ (*Mb*tRNA-CUA) and the acetyl-lysyl-tRNA-sythetase (AcKRS3). AcKRS3 and *Mb*tRNA$_{CUA}$ enable the site-specific incorporation of acetyl-L-lysine into proteins as response to an amber codon. This was done by supplementing the *E. coli* BL21 (DE3) culture with 10 mM N-(e)-acetyl-lysine (Chem-Impex International, Inc.) and 20 mM nicotinamide to inhibit the *E. coli* CobB deacetylase at an OD$_{600}$ of 0.6 (37 °C, 160 rpm). Subsequently, the temperature was lowered to 18 °C and cells were grown for another 30 min at 160 rpm. Protein expression was induced by addition of 200 μM IPTG and protein expression conducted overnight (18 °C, 160 rpm). The cells were harvested by centrifugation (4000 × *g*, 20 min) and resuspended in buffer A. Cell lysis and protein purification was performed as described above.

**Histone western blot**. HeLa cells were lysed by boiling in 2% lithium dodecyl sulfate (LDS) (1x NuPAGE LDS Sample Buffer, ThermoFischer Scientific), followed by sonication to disrupt genomic DNA. Proteins were separated on a 4-12 NuPAGE gel (ThermoFischer Scientific) and transferred to a nitrocellulose membrane (BioRad). Acetylated proteins were visualized by immunoblot using pan-anti-acetylated lysine antibody (Immunechem #ICP0380) at a 1/1000 dilution and goat anti-rabbit horseradisch peroxidase (HRP) (BioRad, STAR124P) secondary antibody at a 1/5000 dilution.

**Data analysis**. Pearson's correlation (*r*) and Wilcoxon tests were performed in R. UniProt keyword analysis was performed using AGOTOOL[10] with an uncorrected *P*-value cutoff of 0.05. Word clouds were generated in R. To estimate histone acetylation copy numbers we used the copy number estimate for histone H4 (5e[7] per cell) for all four core histones. This is because copy number estimates for histones are complicated by the presence of multiple isoforms of H2B, H2A, and H3. The copy number estimate for H4 is consistent with a previous analysis that estimated that core histones represent ~4% of the total protein in HeLa cells, and with the DNA content of HeLa cells[41]. We furthermore used the median stoichiometry for sites that occurred on multiple histone isoforms (such as H2B K5) to avoid over-counting these sites.

**Reporting summary**. Further information on experimental design is available in the Nature Research Reporting Summary linked to this article.

## Data availability
The raw mass spectrometry data have been deposited to the ProteomeXchange Consortium via the PRIDE[40] partner repository with the dataset identifier PXD009994. The source data underlying Figs. 1a, 2a-b, 2d-e, 3a-d, 4a, 4c-d, 5a-c and Supplementary Figs. 1a-c, 2a-c, and 3 are provided as a Source Data file. A reporting summary for this Article is available as a Supplementary Information file. All other data supporting the findings of this study are available from the corresponding authors upon reasonable request.

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

## Acknowledgements

We thank the members of our laboratory for their helpful discussions. L.B. and M.L. were funded by the German Research Foundation (CECAD and DFG Research Grant Number LA2984/5-1). C.C. is supported by the Hallas Møller Investigator award (NNF14OC0008541) from the Novo Nordisk. B.T.W. is supported by a grant from the Novo Nordisk Foundation (NNF15OC0017774). The Novo Nordisk Foundation Center for Protein Research is supported financially by the Novo Nordisk Foundation (Grant agreement NNF14CC0001).

## Author contributions

B.T.W. and C.C. designed the research and supervised the project. B.K.H. performed stoichiometry measurements and contributed to data interpretation. R.G. performed validation experiments. L.B and M.L. provided recombinant acetylated proteins. D.L. and T.N. provided bioinformatics support. B.T.W. analyzed data, prepared figures, and wrote the manuscript. All authors read and commented on the manuscript.

## Additional information

**Competing interests:** The authors declare no competing interests.

