## [Peer Review File · Nature Communications]

Reviewers' comments:

Reviewer #1 (Remarks to the Author):

In this manuscript Hansen et al. focus their efforts into studying the acetylation stoichiometry in a human cervical cancer cell line (Hela). A well-developed quantitative proteomics method was utilized, experiments were carefully executed and results were validated carefully. The study set up is elegant, and discussions based on stoichiometry distribution were done thoroughly covering GO analysis, protein consensus motif, histone acetylation, deacetylases/acetyltransferases and functional modified sites, etc.

This work provides valuable resource for studying protein regulation by acetylation in human cells. Still, there are several concerns to be addressed.

1. In the introduction, the authors brought up the concept of enzymatic/nonenzymatic acetylation, one question is that can stoichiometry resolve the unknown enzymatic/nonenzymatic acetylations? Or in another word, does higher stoichiometry necessarily implicate enzymatic or functional acetylation?
2. Although the calculation of acetylation stoichiometry has been explained in the authors' previous studies in other organisms, more detailed descriptive information regarding the calculation equation in this manuscript is required, as well as the experimental set up of PCA and dilutions. Since comparison to 1% chemically acetylated peptides gave high quantification error, how the quantitative dataset was actually processed including different dilutions? One concern is that does chemical acetylation affect tryptic digestion regarding both cleavage sites and efficiency? Such difference in SILAC light and heavy can affect stoichiometry measurement. Moreover, is the paucity of high stoichiometry measured in this study due to the un-accurate measurement of high stoichiometry in the methodology?
3. In Fig.1F, comparing PCA and AQUA, "Furthermore, stoichiometry measurements differed by a factor of two or less for a majority (7/10) of the analyzed peptides". In fact, besides the three sites showing 3.4-, 6-, and 7-fold higher stoichiometry by PCA, another two sites showed 2- fold changes. Together the comparison (5/10) did not strongly support the correlation of precision by two methods. Therefore, I think the authors need to tone down a bit the interpretation.
4. In Page 6 line 21, to support the last sentence regarding validation using recombinant acetylated proteins, it is Fig.1G that should be referred to, not Fig.S1B.

5. There are a few places in the abstract, results and figures where different total numbers of quantified acetylated sites/peptides/proteins were indicated. E.g. 6829 sites in the abstract and Page 8 line 22 while 6753 sites in Fig.2B; 2535 acetylated proteins in the abstract and 2527 Ac proteins in Fig. 2A, etc. Some explanation/description is appreciated.

6. Histone acetylation sites constitute a majority of acetylation sites in cells, so it's important to accurately measure histone acetylation stoichiometry. N-tails of histone proteins, especially a high variety of acetylation on H3 and H4 N-tails are of great importance in gene transcription. H3 and H4 N-tail peptides via tryptic digestions often carry two or more acetylation sites, such as H3K9/K14, H3K18/K23, H3K27/K36 and H4K5/K8/K12/K16, most of which often present high abundances in all modified forms. Based on the criteria of filtering singly acetylated peptides for quantitation, these di-acetylated and multiply-acetylated H3/H4 peptides were excluded, which I think would affect the quantification accuracy of acetylation stoichiometry of histone peptides. In Fig. 4A, only three sites (H3K23, H3K14 and H4K16) were quantified in this study. Moreover, histone proteins are highly enriched in lysine and arginine residues, and direct trypsin digestion can produce too small peptides which are not suitable for C18 column. Nevertheless, missed cleavage due to different Lys/Arg modifications also often occur resulting in histone peptides with more than two Lys. All these makes quantification of histone modification more complicated. Therefore, although those three sites showed similar stoichiometry as compared to four other studies, the authors need to explain more about the compatibility of their strategy with comprehensive analysis of histone acetylation stoichiometry.

7. In Page 11 line 4, to support statement regarding H3K27/K36 and H2B, it is Fig.4A and 4B that should be referred to, not Fig.3A and 3B. CBP/p300 acetyltransferases target H3K27, H3K36 and N-terminal H2B. However, the stoichiometry of H3K27/K36 were not measured in this study. The values by Feller et al and Zheng et al differed in 10-20 fold (0.3%-2.7%/ 0.3-6.9%). Many N terminal H2B sites were measured in this study, however differing in wide ranges from 0.2% to 5.6%. Additionally, CBP/p300S also targets H3K14 which showed more than 15% stoichiometry in this study and even higher in two other studies (Fig. 4A and 4B). So, how to conclude that "the stoichiometry of these sites is comparable, suggesting that CBP/p300 acetylates these positions to a similar degree?"

8. For Fig. 4C, the molecular weights of histone proteins better be included either in the figure or in the text for a clear and conclusive indication.

9. Based on the result and discussion about stoichiometry of functionally characterized sites, it seems like high stoichiometry acetylation tends to relate to functional sites. In Page 11 Line 9-10, "Here we find that H2BK120 is not more than 0.02% acetylated in HeLa cells, further suggesting that this site is not targeted by acetyltransferases." Why so sure that low stoichiometry acetylation sites are not targeted by enzymes to perform any function? And what is the range/cut off of high or low stoichiometry regarding functional acetylation? In Page 10, when analyzing using IceLogo, the author commented that "cysteine residues may promote nonenzymatic acetylation of downstream lysine residues, and that these sites constitute a substantial portion of high (>0.23%) stoichiometry acetylation". >0.23% is high stoichiometry acetylation, however they implicated non-functional sites. The authors need to explain more about the range of stoichiometry acetylation and their functional meanings. Moreover, in Fig.5A, only fraction high (>1%) stoichiometry was analyzed and compared regarding deacetylases-regulated sites. So, why are only these high stoichiometry acetylation sites of greater interests and chosen for enzyme analysis?

10. In Materials and Methods, the order of methods for recombinant acetylated proteins and AQUA peptides should be inverted considering the order of reporting the results in two validations.

Reviewer #2 (Remarks to the Author):

This paper reports on a carefully executed study of protein acetylation events using an optimized proteomics approach. This work extends previous work from the same group (refs 6-8) to the HeLa cell proteome/acetylome.

The study is of very high technical quality. The proteomics approach based on serial dilution SILAC is complemented by select other techniques to establish stoichiometry levels of proteins. The method will be of interest to biologists and proteomics researchers, but the principles were already published before (refs 6-8).

The biological findings of the study are interesting, as they challenge previous reports on protein acetylation. The present work finds that protein acetylation levels are in general very low. Histones are a special class of proteins that are highly acetylated in a site-specific manner.

The main concern about the biological findings is that they are obtained from HeLa cells, which are transformed human cells with high levels of aneuploidy, multiple copies of chromosomes. Most of

the potentially interesting findings of the present study are in the context of nuclear proteins, and so the conclusions may not hold for healthy cells and tissues. This reduces my enthusiasms for this work.

Reviewers' comments:

Reviewer #1 (Remarks to the Author):

In this manuscript Hansen et al. focus their efforts into studying the acetylation stoichiometry in a human cervical cancer cell line (Hela). A well-developed quantitative proteomics method was utilized, experiments were carefully executed and results were validated carefully. The study set up is elegant, and discussions based on stoichiometry distribution were done thoroughly covering GO analysis, protein consensus motif, histone acetylation, deacetylases/acetyltransferases and functional modified sites, etc. This work provides valuable resource for studying protein regulation by acetylation in human cells. Still, there are several concerns to be addressed.

Thank you for your kind remarks.

1. In the introduction, the authors brought up the concept of enzymatic/nonenzymatic acetylation, one question is that can stoichiometry resolve the unknown enzymatic/nonenzymatic acetylations? Or in another word, does higher stoichiometry necessarily implicate enzymatic or functional acetylation?

This is an important point to address and has been added to the discussion section

“High stoichiometry sites may be particularly interesting because they are more likely to be enzyme-catalyzed. However, more studies are required to determine if high stoichiometry is a good indicator of functional importance. Enzymatic acetylation does not necessarily indicate a regulatory function, and high stoichiometry acetylation may also occur by nonenzymatic mechanisms.”

2. Although the calculation of acetylation stoichiometry has been explained in the authors' previous studies in other organisms, more detailed descriptive information regarding the calculation equation in this manuscript is required, as well as the experimental set up of PCA and dilutions. Since comparison to 1% chemically acetylated peptides gave high quantification error, how the quantitative dataset was actually processed including different dilutions?

These details were provided in the Methods section, as follows.

*“4. Stoichiometry was calculated as follows; Stoichiometry (S), degree partial chemical acetylation (C), dilution factor for acetylated peptides (D), and SILAC ratio partial chemical acetylation /native acetylation (R). $S=(C)/((R*D)-(1-C))$. The dilution factors (D) were as follows:*

Experiment 1, (~1% D=4.23), (~0.1% D=42.3), (~0.01% D=423), Experiment 2, (~1% D=6.37), (~0.1% D=63.7), (~0.01% D=637)."

Additional details are provided in the revised manuscript to clarify the degree of partial chemical acetylation (which was only presented in Figure 1A and Supplementary Figure 1A in the original submission).

"The median degree of partial chemical acetylation (C) was 3.53% and 10.38%, for experiments 1 and 2, respectively, (Fig. S1A)."

We also included the formulas used to calculate stoichiometry in the supplementary Excel table (Supplementary Table 1a), so that these calculations are accessible for each acetylated peptide.

One concern is that does chemical acetylation affect tryptic digestion regarding both cleavage sites and efficiency? Such difference in SILAC light and heavy can affect stoichiometry measurement.

The modest impact of partial chemical acetylation on tryptic digestion is shown in Fig. S1A and differs depending on whether the peptide is generated by cleaving 2 lysine residues, 1 lysine residue, or 0 lysines (no effect). Due to the relatively low degree of partial chemical acetylation tryptic cleavage efficiency is only modestly reduced, even for peptides generated by cleaving at 2 lysine residues. Although we did not correct for these reductions, the difference in calculated stoichiometry would be modest.

Moreover, is the paucity of high stoichiometry measured in this study due to the un-accurate measurement of high stoichiometry in the methodology?

Short answer: No

Long answer: The un-accurate measurement of high stoichiometry acetylation is a product of the small differences that would occur between PCA peptides and native acetylated peptides. For example, 5% PCA at a site with 50% native stoichiometry would result in a PCA peptide at 52.5% stoichiometry, and a SILAC ratio of just 1.05. Whether or not we accurately measure the high stoichiometry has no impact on finding these peptides. High stoichiometry peptides would occur as peptides with SILAC ratios that are less than ~1 (at 1% PCA sites with >1% stoichiometry would have SILAC ratios less than 1.19). In experiment 1, only 203 (3.3%) peptides quantified at high PCA (~1%) had SILAC ratios <1. We calculated stoichiometry for 66 of these peptides and another 61 peptides were disqualified because the SILAC ratio did not agree with the SILAC ratio measured at a different concentration of PCA (~0.1% and/or ~0.01%).

76 peptides were only quantified at ~1% PCA and we were unable to assess the accuracy of their SILAC ratios. Of the 128 peptides whose accuracy we could assess, 57 (~45%) were accurately measured. Therefore, it is likely that ~45% of these 76 peptides (34) would harbor high stoichiometry acetylation. This would increase the number of high stoichiometry peptides in our data (from experiment 1) by ~150% (60+34). However, to be fair we should also consider how many peptides would be added by including similar peptides at a different concentration of PCA. 6,864 peptides were only quantified at a PCA of 0.1%, if we similarly extrapolated from the error rate (~35%) at this concentration of PCA, we would add ~4,461 peptides, of which just 0.5% had SILAC ratios <1. Thus, these data would mainly add low stoichiometry acetylation, and the frequency (~1%) of high stoichiometry (>1%) acetylation reported in our dataset would not be altered. Our data does not suggest that we are systematically underestimating the number of high stoichiometry acetylation sites, or that we are failing to detect these sites.

3. In Fig.1F, comparing PCA and AQUA, “Furthermore, stoichiometry measurements differed by a factor of two or less for a majority (7/10) of the analyzed peptides”. In fact, besides the three sites showing 3.4-, 6-, and 7-fold higher stoichiometry by PCA, another two sites showed 2- fold changes. Together the comparison (5/10) did not strongly support the correlation of precision by two methods. Therefore, I think the authors need to tone down a bit the interpretation.

As quoted above, we claimed that 7/10 sites differed by a factor of “two or less”, not “less than 2”. Thus, the claim of 7/10 sites is accurate. Regardless, whether or not these data strongly support the precision of our method is somewhat subjective. There is no precedent for the precision of stoichiometry measurements, as so few studies have validated their measurements. Unless both methods are systematically biased in the same manner, our data suggests that most of our stoichiometry measurements are within a factor of two from the actual value (likely less, since the imprecision of AQUA measurements also contributes to this variability), which I would consider very remarkable precision for this type of measurement made in a global manner. To acknowledge the subjective nature of this interpretation we revised the manuscript as follows

“We think that the agreement between these two methods is notable when considering all possible sources of variability in each measurement.”

4. In Page 6 line 21, to support the last sentence regarding validation using recombinant acetylated proteins, it is Fig.1G that should be referred to, not Fig.S1B.

Thank you, our mistake.

5. There are a few places in the abstract, results and figures where different total numbers of

quantified acetylated sites/peptides/proteins were indicated. E.g. 6829 sites in the abstract and Page 8 line 22 while 6753 sites in Fig.2B; 2535 acetylated proteins in the abstract and 2527 Ac proteins in Fig. 2A, etc. Some explanation/description is appreciated.

The difference between the number of acetylated proteins reported in the abstract and in Fig.2 is because we were unable to obtain copy number estimates for every acetylated protein, thus comparisons with protein copy numbers were limited to the 2488 proteins for which we had copy number estimates (as reported correctly in Fig. 2B). The number of acetylated proteins reported in Fig.2A is not correct, and has been revised to 2488. The incorrect number was carried over from a previous analysis and were not updated (although the figure was updated).

6. Histone acetylation sites constitute a majority of acetylation sites in cells, so it's important to accurately measure histone acetylation stoichiometry. N-tails of histone proteins, especially a high variety of acetylation on H3 and H4 N-tails are of great importance in gene transcription. H3 and H4 N-tail peptides via tryptic digestions often carry two or more acetylation sites, such as H3K9/K14, H3K18/K23, H3K27/K36 and H4K5/K8/K12/K16, most of which often present high abundances in all modified forms. Based on the criteria of filtering singly acetylated peptides for quantitation, these di-acetylated and multiply-acetylated H3/H4 peptides were excluded, which I think would affect the quantification accuracy of acetylation stoichiometry of histone peptides. In Fig. 4A, only three sites (H3K23, H3K14 and H4K16) were quantified in this study. Moreover, histone proteins are highly enriched in lysine and arginine residues, and direct trypsin digestion can produce too small peptides which are not suitable for C18 column. Nevertheless, missed cleavage due to different Lys/Arg modifications also often occur resulting in histone peptides with more than two Lys. All these makes quantification of histone modification more complicated. Therefore, although those three sites showed similar stoichiometry as compared to four other studies, the authors need to explain more about the compatibility of their strategy with comprehensive analysis of histone acetylation stoichiometry.

We agree with the points raised by the reviewer and have revised the manuscript to better indicate the limitations of our methodology with respect to histone sites. The following text was added to the results section "Histone Acetylation"

"However, our method does not measure the stoichiometry of doubly acetylated peptides that can arise from the lysine-rich N-terminal tails of the core histones H2A, H2B, H3, and H4. Feller et al¹⁸ found that the stoichiometry of mono-acetylated histones H3 and H4 were more abundant than di-acetylated H3 and H4, with the exception of H4 K5+K12 and H3 K18+23 (which is more abundant than K18 alone, but less abundant than K23 alone). We detected doubly acetylated peptides containing the following sites on H2A (K5+K9, K9+K11, K11+K13), H2B (K5+K11, K11+K12, K15+K16, K16+K20, K20+K23, K34+K43, and K116+K120), H3 (K9+K14, K18+K23, and K27+K36), and H4 (K8+K12 and K12+K16). The abundance of di-acetylated

peptides remains unexplored for H2B, and these peptides may be more abundant than their mono-acetylated counterparts used to calculate stoichiometry in this study. In addition, some acetylated peptides from histone tails may not be detected because of their small size. Thus, our estimates of histone acetylation stoichiometry may underestimate the actual native stoichiometry at these positions because these sites also occur on di-acetylated peptides or peptides that we are unable to detect with our methodology.”

7. In Page 11 line 4, to support statement regarding H3K27/K36 and H2B, it is Fig.4A and 4B that should be referred to, not Fig.3A and 3B.

Thank you, this is revised

CBP/p300 acetyltransferases target H3K27, H3K36 and N-terminal H2B. However, the stoichiometry of H3K27/K36 were not measured in this study. The values by Feller et al and Zheng et al differed in 10-20 fold (0.3%-2.7%/ 0.3-6.9%). Many N terminal H2B sites were measured in this study, however differing in wide ranges from 0.2% to 5.6%. Additionally, CBP/p300S also targets H3K14 which showed more than 15% stoichiometry in this study and even higher in two other studies (Fig. 4A and 4B). So, how to conclude that “the stoichiometry of these sites is comparable, suggesting that CBP/p300 acetylates these positions to a similar degree?

We agree with the reviews concerns and removed the claim

“Notably, the stoichiometry of these sites is comparable (Fig. 4A and 4B), suggesting that CBP/p300 acetylates these positions to a similar degree.”

As an aside, we found no regulation of H3K14 in our analysis of CBP/p300 regulated sites using a combination of genetic KO and two highly specific chemical inhibitors (PMID: 29804834)

8. For Fig. 4C, the molecular weights of histone proteins better be included either in the figure or in the text for a clear and conclusive indication.

The positions of the Histone forms and their MW have been indicated in the figure panel.

9. Based on the result and discussion about stoichiometry of functionally characterized sites, it seems like high stoichiometry acetylation tends to relate to functional sites. In Page 11 Line 9-10, “Here we find that H2BK120 is not more than 0.02% acetylated in HeLa cells, further suggesting that this site is not targeted by acetyltransferases.” Why so sure that low stoichiometry acetylation sites are not targeted by enzymes to perform any function?

We agree with the reviewer that we cannot rule out enzymatic acetylation or function at this position, and we removed the following comments from the results section.

“Histone H2B K120 is also a reported target of CBP/p300²¹. However, H2B K120 is unaffected by CBP/p300 knock-out or catalytic inhibition, and H2B K120ac antibody may cross-react with other sites¹². Here we find that H2BK120 is not more than 0.02% acetylated in HeLa cells, further suggesting that this site is not targeted by acetyltransferases.”

And what is the range/cut off of high or low stoichiometry regarding functional acetylation? In Page 10, when analyzing using IceLogo, the author commented that “cysteine residues may promote nonenzymatic acetylation of downstream lysine residues, and that these sites constitute a substantial portion of high (>0.23%) stoichiometry acetylation”. >0.23% is high stoichiometry acetylation, however they implicated non-functional sites. The authors need to explain more about the range of stoichiometry acetylation and their functional meanings. It was inaccurate to refer to these sites as “high” stoichiometry acetylation in that sentence and was an oversight on our part. However, these sites were accurately described in the preceding sentences...

“Cysteine residues were notably overrepresented for sites with >0.23% stoichiometry (10-fold higher than median stoichiometry), particularly in the -4, -3, and -2 positions (Fig. 3D). However, this bias was absent when examining sites with >1% stoichiometry, indicating that this overrepresentation was associated with sites with moderately elevated stoichiometry.”

To address the reviewer's concern we revised the sentence in question to

“cysteine residues may promote nonenzymatic acetylation of downstream lysine residues, and these sites constitute a substantial portion of sites with an elevated (>0.23%) stoichiometry of acetylation.”

Moreover, in Fig.5A, only fraction high (>1%) stoichiometry was analyzed and compared regarding deacetylase-regulated sites. So, why are only these high stoichiometry acetylation sites of greater interest and chosen for enzyme analysis?

As noted above, we mistakenly referred to >0.23% (10x higher than median) as high stoichiometry. Although, these classifications are, by nature, arbitrary. However, we thank the reviewer for raising this point, since the trend and bias we observed for >1% stoichiometry is also observed at >0.23% stoichiometry, and we believe this strengthens our findings. Therefore, the figure has been updated to include these data and the text has been revised.

10. In Materials and Methods, the order of methods for recombinant acetylated proteins and

AQUA peptides should be inverted considering the order of reporting the results in two validations.

Done

Reviewer #2 (Remarks to the Author):

This paper reports on a carefully executed study of protein acetylation events using an optimized proteomics approach. This work extends previous work from the same group (refs 6-8) to the HeLa cell proteome/acetylome.

The study is of very high technical quality. The proteomics approach based on serial dilution SILAC is complemented by select other techniques to establish stoichiometry levels of proteins. The method will be of interest to biologists and proteomics researchers, but the principles were already published before (refs 6-8).

The biological findings of the study are interesting, as they challenge previous reports on protein acetylation. The present work finds that protein acetylation levels are in general very low. Histones are a special class of proteins that are highly acetylated in a site-specific manner.

The main concern about the biological findings is that they are obtained from HeLa cells, which are transformed human cells with high levels of aneuploidy, multiple copies of chromosomes. Most of the potentially interesting findings of the present study are in the context of nuclear proteins, and so the conclusions may not hold for healthy cells and tissues. This reduces my enthusiasms for this work.

We understand the reviewer's concern regarding the chosen cell type. We previously determined stoichiometry in mouse liver tissue (PMID: 26358839), which partly addresses this concern. However, we do not consider our findings in HeLa cells to be uninformative. HeLa cells are one of the world's most-studied cell line, thus our findings are relevant for those that use this cell line as a model for their experiments. Furthermore, there is no evidence to suggest that acetylation levels are grossly altered in HeLa or other cancer-derived cell lines that are commonly used in research.

REVIEWERS' COMMENTS:

Reviewer #1 (Remarks to the Author):

The rebuttal letter was well prepared. The responses addressed all my concerns.